# Garcinol Attenuates Lipoprotein(a)-Induced Oxidative Stress and Inflammatory Cytokine Production in Ventricular Cardiomyocyte through α7-Nicotinic Acetylcholine Receptor-Mediated Inhibition of the p38 MAPK and NF-κB Signaling Pathways

**DOI:** 10.3390/antiox10030461

**Published:** 2021-03-16

**Authors:** Nen-Chung Chang, Chi-Tai Yeh, Yen-Kuang Lin, Kuang-Tai Kuo, Iat-Hang Fong, Nicholas G. Kounis, Patrick Hu, Ming-Yow Hung

**Affiliations:** 1Division of Cardiology, Department of Internal Medicine, Taipei Medical University Hospital, Taipei 110, Taiwan; 2Division of Cardiology, Department of Internal Medicine, School of Medicine, College of Medicine, Taipei Medical University, Taipei 110, Taiwan; ncchang@tmu.edu.tw; 3Taipei Heart Institute, Taipei Medical University, Taipei 110, Taiwan; 4Department of Medical Research and Education, Taipei Medical University—Shuang Ho Hospital, New Taipei City 23561, Taiwan; ctyeh@s.tmu.edu.tw (C.-T.Y.); impossiblewasnothing@hotmail.com (I.-H.F.); 5Department of Medical Laboratory Science and Biotechnology, Yuanpei University of Medical Technology, Hsinchu City 30015, Taiwan; 6Biostatistics Center, Office of Data Science, Taipei Medical University, Taipei 110, Taiwan; 7Graduate Institute of Data Science, Taipei Medical University, Taipei 110, Taiwan; 8Research Center of Big Data, College of Management, Taipei Medical University, Taipei 110, Taiwan; robbinlin@tmu.edu.tw; 9Division of Thoracic Surgery, Department of Surgery, Shuang Ho Hospital, Taipei Medical University, New Taipei City 23561, Taiwan; doc2738h@gmail.com; 10Division of Thoracic Surgery, Department of Surgery, School of Medicine, College of Medicine, Taipei Medical University, Taipei 110, Taiwan; 11Department of Internal Medicine, Division of Cardiology, University of Patras Medical School, 26221 Patras, Greece; ngkounis@otenet.gr; 12Department of Cardiology, University of California, Riverside, CA 92521, USA; patrick.p.hu@gmail.com; 13Department of Cardiology, Riverside Medical Clinic, Riverside, CA 92506, USA; 14Division of Cardiology, Department of Internal Medicine, Shuang Ho Hospital, Taipei Medical University, New Taipei City 23561, Taiwan

**Keywords:** garcinol, nicotinic receptor, α7-nAChR, NF-κB signaling

## Abstract

Garcinol, a nicotinic acetylcholine receptor (nAChR) antagonist, has recently been established as an anti-inflammation agent. However, the molecular mechanism by which garcinol suppresses inflammation in the context of acute myocardial infarction (AMI) remains unclear. **Hypothesis:** We hypothesized that the administration of physiological doses of garcinol in mice with isoproterenol-induced AMI decreased the effect of lipoprotein(a) (Lp(a))-induced inflammation both in vivo and in vitro via the α7-nAChRs mediated p38 mitogen-activated protein kinase (MAPK)/nuclear factor kappa-light-chain-enhancer of activated B cells (NF-kB) signaling pathway. We analyzed altered reactive oxygen species (ROS) generation, the production of superoxide by mitochondria, cytokine expression patterns, and the role of the p38 MAPK/NF-κB signaling pathway after Lp(a)-stimulated human ventricular cardiomyocyte AC16 cells were treated with increasing doses of garcinol. C-reactive protein (CRP), interleukin (IL)-1β, IL-6, or tumor necrosis factor (TNF)-α production were detected by enzyme-linked immunosorbent assay. The Cell Counting Kit-8 assay was used to evaluate drug cytotoxicity. Western blots and confocal fluorescence microscopy were used to determine altered expression patterns of inflammatory biomarkers. We also examined whether the therapeutic effect of garcinol in AMI was mediated in part by α7-nAChR. Lp(a)-induced inflammatory cardiomyocytes had increased expression of membrane-bound α7-nAChRs in vitro and in vivo. Low-dose garcinol did not affect cardiomyocyte viability but significantly reduced mitochondrial ROS, CRP, IL-1β, IL-6, and TNF-α production in Lp(a)-stimulated cardiomyocytes (*p* < 0.05). The Lp(a)-induced phosphorylation of p38 MAPKs, CamKII, and NFκB, as well as NFκB-p65 nuclear translocation, was also suppressed (*p* < 0.05) by garcinol, while the inhibition of p38 MAPK by the inhibitor SB203580 decreased the phosphorylation of extracellular signal-regulated kinase (ERK) and p38 MAPK. Garcinol protected cardiomyocytes by inhibiting apoptosis and inflammation in mice with AMI. Furthermore, garcinol also enhanced the expression of microRNA-205 that suppressed the α7-nAChR-induced p38 MAPK/NF-κB signaling pathway. Garcinol suppresses Lp(a)-induced oxidative stress and inflammatory cytokines by α7-nAChR-mediated inhibition of p38 MAPK/NF-κB signaling in cardiomyocyte AC16 cells and isoproterenol-induced AMI mice.

## 1. Introduction

Acute myocardial infarction (AMI) with the subsequent cardiomyocyte apoptosis and adverse left ventricular remodeling constitutes the structural basis for ischemic heart failure [1], during which stressed cardiomyocytes due to hypoxia and ischemia can trigger the inflammatory response [2]. Restoration of blood flow may further augment tissue damage via reperfusion injury due to abrupt re-oxygenation and reactive oxygen species (ROS) [2]. Although timely coronary reperfusion is the most effective way of limiting myocardial injury, therapeutic manipulation of the ensuing tissue inflammation and its subsequent active suppression and resolution, remains elusive [2].

Lp(a) is a genetically determined causal factor for AMI [3]. While the risk of AMI increased with increasing Lp(a) concentrations, which is independent of Lp(a) isoform size, Lp(a) concentration and isoform size varied markedly between ethnic groups [4]. Higher Lp(a) concentrations carried an especially high population burden in South Asians [4,5]. Lp(a) is highly prone to oxidative modifications and leads to extensive formation of pro-inflammatory and pro-atherogenic oxidized phospholipids, oxysterols, oxidized lipid-protein adducts that lead to atherosclerotic progression [6,7]. Notably, oxidative stress-mediated actions are considered to be one of the most critical factors responsible for coronary artery atherosclerosis [8] and spasm [9]. Mechanistically, Lp(a) stimulates and enhances the migration and proliferation of cells through activating Src family kinases and mitogen-activated protein kinases (MAPK), including the extracellular signal-regulated kinase (ERK), p38, and c-Jun N-terminal Kinases [10]. It also increases the expression of adhesion molecules such as intercellular adhesion molecule 1 (ICAM-1, CD54) and E-selectin (CD62E) [11]. However, the underlying mechanisms by which Lp(a) mediates post-infarct inflammation, and contributes to post-infarct cardiomyocyte apoptosis, are incompletely understood.

Among natural dietary plant-derived phytochemicals that have been studied dramatically in the last few decades with the aim to protect from a number of chronic diseases, garcinol revealed its potential therapeutic effects in in vitro studies, such as its anti-oxidative, anti-inflammatory, and anti-cancer properties [12]. While the antioxidant nature of garcinol largely depends on its polyphenolic structure, the anti-inflammatory activities of garcinol involve modulating the expression of pro-inflammatory cytokines (e.g., interleukin (IL)-1β, IL-6), cyclooxygenase-2, inducible nitric oxide synthetase downregulating signal transducers and activators of transcription-3 (STAT-3), and inhibiting nuclear factor kappa-light-chain-enhancer of activated B cells (NF-kB), microRNAs (miRNAs), and vimentin [13,14]. Specifically, garcinol at very low concentration (1 μM) inhibits nicotinic acetylcholine receptor (nAChR) [15,16], among which the downregulation of the α9-nAChR attenuates nicotine-induced human breast cancer cell proliferation [14]. On the other hand, α7-nAChR is important in regulating the production of pro-inflammatory cytokines [17], such as tumor necrosis factor (TNF)-α, IL-6, [18], and C-reactive protein (CRP) [19]. Additionally, α7-nAChR recycling is mediated by Ca^2+^/calmodulin-dependent kinase II (CaMKII) [20]. Furthermore, while the inflammation and oxidative stress in AMI has been related to Ras-homologous (Rho)A guanosine triphosphate hydrolase enzymes (GTPase) and downstream effector Rho-kinase (ROCK1, ROCK2) pathway [21,22], α7-nAChR is involved in the activation of the RhoA-GTP (guanosine triphosphate)/ROCK pathway [23]. Recent studies have shown that NF-κB is cardioprotective during acute hypoxia and reperfusion injury [24]. However, prolonged activation of NF-κB appears to be detrimental and promotes heart failure by eliciting signals that trigger chronic inflammation through enhanced elaboration of cytokines including TNF-α and IL-6, leading to apoptosis [24]. The underlying mechanisms that account for the diverse outcomes of NF-κB on cardiomyocyte fate remain to be elucidated. Among CMGC kinase family (cyclin-dependent kinases, MAPKs, glycogen synthase kinases (GSKs), and cyclin-dependent kinase-like kinases), GSK-3β appears to be the more influential kinase in terms of both pro- and anti-inflammatory actions [25]. Its specific effect, however, appears to depend on the respective cellular and molecular conditions [25]. Taken together, it remains unknown whether the anti-inflammatory effects of garcinol could be mediated through α7-nAChR and downstream signaling mechanisms in myocardial injury.

While AMI is a multifactorial and progressive disease that has not yet been fully understood, studies on isoproterenol-induced cardiotoxicity mouse models provide a good insight into this disease and clearly indicate the involvement of inflammation and apoptosis. Thus, the purposes of this study were to investigate (1) the toxicity of Lp(a) on cardiomyocytes in in vitro AC16 cells and in vivo isoproterenol-induced AMI mice, (2) the role of α7-nAChR and downstream signaling mechanisms in regulating Lp(a)-induced cardiomyocyte apoptosis and inflammation, (3) the protective effects of garcinol on α7-nAChR and downstream signaling mechanisms, (4) the effects of Lp(a) and garcinol on filamentous actin (F-actin), which is essential for mobility and contraction of cells, and (5) the mechanism by which garcinol modulated α7-nAChR functions.

## 2. Materials and Methods

### 2.1. Drugs, Cell Line and Culture

Garcinol (sc-200891, HPLC purity ≥95%) purchased from Santa Cruz Biotechnology (Santa Cruz, CA, USA) and SB203580 (a p38 MAPK inhibitor), purchased from Sigma. Dissolved in dimethyl sulfoxide to prepare a 10 mM stock and stored at −20 °C. As per the requirement for different assays, the stock was further diluted using growth media. The human AC16 Human Ventricular Cardiomyocyte cell line was purchased from American Type Cell Culture. The cells were maintained in Dulbecco’s Modified Eagle Medium/Nutrient Mixture F-12 supplemented with 10% fetal bovine serum and 1% penicillin/streptomycin (Invitrogen, Life Technologies, Carlsbad, CA, USA) and incubated in 5% humidified CO_2_ incubator at 37 °C. Media was changed every 48–72 h, and the cells were sub-cultured when they reach 80–90% confluency.

### 2.2. Cell Counting Kit-8 Cell Viability Assay

A Cell Counting Kit-8 assay (Dojindo Laboratories, Kumamoto, Japan) were used to measure the survival/proliferation of AC16 cells as per the manufacturer’s protocol.

### 2.3. Quantitative Real-Time Reverse Transcriptase-Polymerase Chain Reaction

Total RNA extracted from ventricular cardiomyocyte AC16 cells treated with Lp(a) or Lp(a) + garcinol using TRIzol-based protocol (Life Technologies), following the manufacturer’s recommendation. A total of 2 ug RNA was reverse transcribed by QIAGEN OneStep real-time polymerase chain reaction (RT-PCR) Kit (QIAGEN, Taiwan), and the PCR was performed under the following condition: reverse transcription at 42 °C for 60 min, amplification for 30 cycles at 94 °C for 30 s, 58 °C for 50 s, and 72 °C for 50 s. The primers sequence for target genes and the microRNA was shown in Appendix A

### 2.4. Detection of Reactive Oxygen Species (ROS) Production

The cell-permeant fluorescent probe 20, 70-dichlorodihydrofluorescein diacetate (DCFH-DA) (Molecular Probes, Eugene, OR, USA) was used to quantify ROS levels in cells. In brief, DCFH-DA is diffused into cells and hydrolysed by intra-cellular esterases, and then, the non-fluorescent 20, 70-dichlorodihydrofluorescein (DCFH) rapidly converted to fluorescent 20, 70-dichlorodifluorescein (DCF) by various ROS. The fluorescence intensity is proportional to the ROS levels within the cell cytosol. Ventricular cardiomyocyte AC16 cells were incubated with DCFH-DA (5 μM) for 30 min at 37 °C in the dark, and the fluorescence signal of 20, 7-dichlorofluorescein (Ex¼490 nm), the oxidation product of DCFH-DA, was excited at 488 nm and observed at 530 nm using confocal laser scanning microscopic analysis and quantified by FACStar flow cytometer. The concentration of nitrate was measured using the nitrate No.780001 (Cayman Chemicals, MI, USA). The lower limit of detection was 2.0 μM.

### 2.5. Detection of Apoptosis by Flow Cytometry

Ventricular cardiomyocyte AC16 cells were seeded into six-well plates, cultured in Dulbecco’s Modified Eagle Medium/Nutrient Mixture F-12 supplemented with 10% fetal bovine serum, and incubated at 37 °C in 5% CO_2_ for 24 h. After that, the medium was replaced with a medium containing Lp(a) (1–10 μM) or garcinol (0.5–2.5 μM) for 24 or 48 h. Cell apoptosis was assayed by using a PE Annexin V Apoptosis Detection Kit I (BD Biosciences) according to the manufacturer’s instructions. Cells were trypsinized by 0.25% trypsin-ethylenediaminetetraacetic acid (EDTA) solution, washed twice with cold PBS, and stained with annexin V-PE (5 μL) and 7-AAD (5 μL) in binding buffer. After incubation at room temperature for 15 min, cell apoptosis was analyzed by BD FACSAria^TM^ III flow cytometer.

### 2.6. Mitochondrial Superoxide Staining Assay

MitoSOX^TM^ Red superoxide indicator (Invitrogen) is a fluorogenic dye that is selective for mtO2^•−^ in live cells. It localizes into cellular mitochondria and is readily oxidized by superoxide, but not other sources of ROS or nitrogen species. The oxidation of the probe is prevented by superoxide dismutase and exhibits a bright red fluorescence upon binding to nucleic acids (excitation=emission maxima ¼ 510 = 580 nm). After Lp(a) exposure, ventricular cardiomyocyte AC16 cells were incubated with MitoSOX^TM^ Red (3 μM) for 10 min at 37 °C as indicated by the manufacturer’s instructions. The AC16 cells were collected by trypsinization and washed in phosphate-buffered saline (PBS) supplemented with 2% fetal bovine serum (FBS). AC16 cells were fixed in 2% paraformaldehyde and suspended in PBS. Measurements were performed in duplicates using the BD flow cytometer (BD Biosciences). MitoSOX^TM^ Red was excited at 488 nm, and the data were collected by a 575 = 26 nm (FL2) channel. The data were presented by histograms in terms of the mean intensity of MitoSOX^TM^ fluorescence normalized to those of the garcinol controls.

### 2.7. Western Blot Analysis

Total proteins of ventricular cardiomyocyte AC16 cells were extracted after treatment from different experiments were separated using the sodium dodecyl sulfate polyacrylamide gel electrophoresis (SDS-PAGE) using Mini-Protean III system (Bio-Rad, Taiwan) and transferred onto polyvinylidene fluoride membranes using Trans-Blot Turbo Transfer System (Bio-Rad, Taiwan). Membranes were incubated overnight at 4 °C in primary antibodies enlisted in Appendix A. Secondary antibodies were purchased from Santa Cruz Biotechnology (Santa Cruz, CA, USA), and an enhanced chemiluminescence (ECL) detection kit was used for the detection of the protein of interests. Images were captured and analyzed using UVP BioDoc-It system (Upland, CA, USA). The Western blot antibodies were shown in Appendix A.

### 2.8. Cell-Based Enzyme-Linked Immunosorbent Assay (ELISA) Analysis

The protocol used for cell enzyme-linked immunosorbent assay (ELISA) of IL-6, TNF-α, CRP, NFĸB, α7-nAChR, clusterin, endothelin-1 and troponin I have been modified from that of Rothlein et al. [26] and Takami et al. [27]. The optical density of each well was determined using a microplate reader at 450 nm within 30 min.

### 2.9. Morphological Changes and Immunofluorescence

Morphological changes characteristic of apoptosis was determined by 4′,6-diamidino-2-phenylindole (DAPI) staining as per the manufacturer’s protocol (Invitrogen, USA). Briefly, 5 × 10^3^ cells were seeded into 6-well plates containing 1–2 mL medium. After 24–36 h, garcinol was added and incubated for another 48 h. Cells were harvested by trypsinization, washed with PBS, and subsequently incubated for 30 min with DAPI at room temperature in the dark for 30 min. Before microscopic analysis, the cells were stained with Prolong Gold Antifade reagent and visualized under a fluorescence microscope (Nikon Eclipse, 80i) with an excitation maximum at 358 nm and an emission maximum at 461 nm. Immunofluorescence, for F-actin staining, cells were incubated with either rhodamine-phalloidin (red fluorescence) or fluorescein phalloidin (green fluorescence; Invitrogen). For the staining of nuclei, sections and/or cells were incubated with 50 μg/mL DAPI in PBS, and then mounted with antifade mounting medium (0.1 M Tris, pH 9.0).

### 2.10. Transfection of miRNA and Anti-miRNA

Transfection of miRNA-205 was performed using the TransIT-X2 system for 12 h. The miRNA-205 sequence is 5′-CAAGUGCUCAG AUGUCUGUGGU-3′. AC16 cells were transfected with a final concentration of 10 nM miRNA-205 according to the manufacturer’s instructions. Anti-miRNA-205 (10 nM) was used to inhibit the expression of endogenous miRNA-205. After transfection of miRNA or anti-miRNA, the media were changed for stabilization, and then garcinol was the induced treatment.

### 2.11. Immunohistochemistry and Terminal Deoxynucleotidyl Transferase Deoxyuridine Triphosphate (dUTP) Nick End Labeling (TUNEL) Assay

Paraffin-embedded heart tissue sections were deparaffinized with xylene, dehydrated with graded alcohol, and incubated with warm deionized water containing 0.3% H_2_O_2_ for 30 min. After endogenous peroxide was eliminated, sections were blocked with serum and added with primary antibodies for incubation at 4 °C overnight. The next day, sections were incubated with IgG antibody-HRP (horseradish peroxidase), and dropwise added with the mixture prepared in the biotin and ABC kit for culture, followed by color development with 3, 3′-diaminobenzidine (DAB) for 10 min. After that, sections were counterstained with hematoxylin, washed, dehydrated, and permeabilized. Lastly, sections were observed using an optical microscope. Transferase dUTP nick end labeling (TUNEL) staining: dewaxed paraffin sections were stained with an In-Situ Cell Death Detection Kit (Roche Diagnostics Corp.) using a previously described protocol [28].

### 2.12. Acute Myocardial Infarction (AMI) Mouse Model Studies

The animal study protocol was approved by the Animal Care and User Committee at Taipei Medical University (Affidavit of Approval of Animal Use Protocol # Taipei Medical University- LAC-2018-0573) consistent with the U.S. National Institutes of Health Guide for the Care and Use of Laboratory Animals. Male C57Bl/6 mice (8-week-old) purchased from BioLASCO (BioLASCO Taiwan Co., Ltd. Taipei, Taiwan). A standard pellet diet and sufficient water were provided to mice and maintained under standard laboratory conditions (21 ± 2 °C; 60–65% humidity) at 12/12 h light and dark cycle in a polycarbonate cage. The induction of AMI was performed in the infarcted group, through subcutaneous administration of isoproterenol at a dose of 10 mg/kg for 3 days and 5 mg/kg for 11 days. Following the isoproterenol was injected, randomly placed into Lp(a) control (Lp(a) 1.5 mg/kg) or garcinol-treated (Lp(a) 1.5 mg/kg, garcinol 1 mg/kg, intraperitoneal (i.p.) injection) group. Post-experiment, the mice were humanely sacrificed, and heart samples were collected for further comparative immunohistochemistry and Western analyses. [29,30].

### 2.13. Statistical Analysis

All experiments were performed at least 3 times in triplicates. All values are presented as the mean ± standard deviation (SD). The comparison between the two groups was made using the 2-sided Student’s *t*-test, while a one-way analysis of variance (ANOVA) was used to compare ≥3 groups. All statistical analyses were performed by SPSS 19.0 software. A *p*-value < 0.05 was considered statistically significant.

## 3. Results

### 3.1. Effect of Lp(a) on Oxidative Stress and α7-nAChR (Nicotinic Acetylcholine Receptor) Mediated Phosphorylation

To examine the effect of Lp(a) on cell viability of human ventricular cardiomyocyte, AC16 were treated with increasing concentration of Lp(a) up to 10 µM. Cell viability of A16 cells (Figure 1A) was measured after 48 h of treatment using a Cell Counting Kit-8 and the morphology of the cells was captured under the microscope (Figure 1B). Lp(a) treatment at 1 µM to 5 µM decreases cell viability to 65%, while treatment with 10 µM of Lp(a) decreased the cell viability to approximately 30% compared to the control group cells (Figure 1A). Apoptosis is a major cause of cell growth inhibition to evaluate the effect of Lp(a) on cell growth. Lp(a) treated and non-treated AC16 cell were stained with annexin V/7-AAD. Flow cytometry analysis of stained cells showed predominant apoptosis of Lp(a) treated cells (Figure 1C). In addition, we quantified the intracellular redox status, and measured DCF fluorescence. These experiments revealed a two-fold increase of fluorescence 24 h post Lp(a) addition in A16 cells. The DCF response was more pronounced in time-dependently increased (3.3-fold). In these cells, Lp(a) increased DCF fluorescence by 5.7-fold (Figure 1D). Nitrate concentrations (surrogate markers for NO production via iNOS) in Lp(a)-treated A16 cells increased by 3-fold (Figure 1E). To estimate the effect α7-nAChR on cardiomyocytes, the protein expression analysis was performed. The result from immunoblotting analysis described in Figure 1F, Lp(a) treatment (10 µM) shows a positive effect on the expression of α7-nAChR and the phosphorylation status of CamKII, p38-MAPK or ERK.

### 3.2. Garcinol Prevents the Apoptotic Cell Death Induced by Lp(a)

To elucidate whether garcinol treatment inhibited cell cytotoxicity caused by Lp(a), we performed cell viability assay by Cell Counting Kit-8 and annexin V/7-AAD apoptosis assay of AC16 cells, sensitized by 1 µM Lp(a), then treated with different concentration of garcinol (0.5–2.5 µM). Different concentrations of garcinol did not affect the viability of cardiomyocytes. Moreover, garcinol inhibited the cell cytotoxicity induced by Lp(a) in concentration-dependent manners, with 0.5 µM being the lowest effective concentration (Figure 2A) and less vulnerable to Lp(a)-related cytotoxic effects, as shown in Figure 2B. Protective effects of garcinol against the cell apoptosis of AC16 cells were analyzed by flow cytometry analysis (Figure 2C), demonstrating that the 24 h incubation of AC16 cells with garcinol (1 μM) significantly reduced the apoptosis induced by Lp(a). Therefore, ROS level was quantified using flow cytometer 6 and 12 h after Lp(a) stimulation (10 μM) with or without garcinol pretreatment (2.5 μM) for 6 h. Similarly, Lp(a)-induced ROS production and mitochondrial superoxide mtO2^•−^ production was markedly decreased by garcinol post-Lp(a) exposure (Figure 2D,E). To further confirm that garcinol was able to prevent α7-nAChR effects on AC16 cells, Western blot and qRT-PCR analysis were used to evaluate the expression of α7-nAChR. Garcinol (0.5–2.5 μM) suppressed α7-nAChR expression both at protein and mRNA level dose-dependently (Figure 2F,G). Collectively, an exposure-response relationship exists between garcinol and the reduction in oxidative stress, in which both the amount and the duration of garcinol used modulate this association.

### 3.3. Garcinol Inhibits the α7-nAChR Mediated Phosphorylation and Expression of Adhesion Molecules in AC16

We determined α7-nAChR mediated phosphorylation and regulation in the expression of adhesion molecules on cardiomyocytes. The protein analysis was performed after garcinol (2.5 µM) treatment. As shown in Figure 3A,B, garcinol inhibited the α7-nAChR along with the inhibition of phosphorylation of key signaling molecules, such as CamKII, ERK and p38 MAPK, reversing the effect caused by Lp(a) treatment. As shown in Figure 3C, Western blot analysis showed that adhesion molecules expression, especially vascular cell adhesion molecule 1 (VCAM-1), ICAM-1, and E-selectin were upregulated by Lp(a) and suppressed by garcinol. The effect of garcinol on reversing Lp(a)-induced increased expression of VCAM-1, ICAM-1, and E-selectin were further demonstrated by mRNA expression analysis showed in Figure 3D, that garcinol treatment significantly reduced the VCAM-1, ICAM-1 and E-selectin expression. To further understand the influence of Lp(a) on the Rho GTPase, we examined the activation of an important Rho GTPases, RhoA-GTP, and its downstream effector, ROCK. Western blot and ROCK activity analysis shows garcinol deactivates the Lp(a)-mediated activated the expression of p-myosin-binding subunit (p-MBS), ROCK1, ROCK2, RhoA-GTP. (Figure 3E,F).

### 3.4. Garcinol Inhibits α7-nAChR/IL-6/NFkB Proinflammatory Response Activation in AC16 Cells

The inhibitory effect of garcinol on the reduction of α7-nAChR-induced activation of the MAPK family and pro-inflammatory response regulators was also evaluated in AC16 cells by ELISA and immunoblotting. In ELISA results as compared to the control group, there was a significant reduction in the level of expression of nicotinic receptor α7-nAChR, along with the expression of proinflammatory cytokines (IL-6, TNF-a, CRP, and NFkB) (Figure 4A). The result, depicted in Figure 4B of protein expression analysis also demonstrated a significant reduction in the NFkB, a transcriptional regulator expression as well as the level of inflammatory factors, such as IL-6, TNF-α and CRP was observed. The significant reduction of mRNA expression in the α7-nAChR-mediated activation and phosphorylation of p38 MAPK-NFkB after 24 h of garcinol (2.5 μM) treatment in AC16 cells were observed (Figure 4C).

### 3.5. Garcinol Prevented Apoptosis and Inhibited Insulin-Like Growth Factor 2 Receptor (IGF2R) through Inactivation of Phosphorylation of Glycogen Synthase Kinase (GSK)-3β/Extracellular Signal-Regulated Kinase (ERK)/p38 Mitogen-Activated Protein Kinase (MAPK) in Cardiomyocyte AC16 Cells

Garcinol inhibited Lp(a)-induced expression of insulin-like growth factor 2 receptor (IGF2R), caspase-3 and p-GSK-3β/NF-κB, and p-ERK/p-p38 MAPK (Figure 5A). SB203580 had a similar inhibitory effect of garcinol. Immunofluorescence analysis of F-actin polymerization using rhodamine-phalloidin dye (F-actin, red-orange), in which nuclei were stained DAPI (blue), showed that the expression of F-actin was increased on garcinol treatment in comparison to Lp(a) and SB203580 treatment (Figure 5B). AC16 cardiomyocytes were treated with condition media with Lp(a) alone and with garcinol (2.5 μM). A substantial reduction in caspase-3 activation, suggesting less apoptosis, was observed in garcinol treatment in comparison to Lp(a) only treatment (Figure 5C). The mir-Target prediction showed the three prime untranslated region (3′UTR) sites of Cholinergic Receptor Nicotinic Alpha 7 Subunit (CHRNA7) was targeted by miR-205 (Figure 5D). Garcinol treatment compared to control significantly upregulated miR-205 expression (Figure 5E).

### 3.6. Garcinol Treatment Suppressed Lp(a) Induced Inflammation and Cell Damage in Isoproterenol-Induced AMI Mice

After the successful induction of AMI in mice (Figure 6A), we investigated the potential of garcinol in reversing the effect of Lp(a) in-vivo. The heart and liver weight of the control group, and Lp(a)-treated mice as compared to the garcinol treatment in mice. Pretreatment with garcinol significantly decreased heart weight and liver weight. (Figure 6B). Hematoxylin and eosin (H&E) staining results showed AMI was alleviated for each group (Figure 6C). These data indicate that garcinol pretreatment inhibited cardiac hypertrophy. Furthermore, the result of TUNEL assay showed that the number of apoptotic cardiomyocytes was increased in the control group and Lp(a)-treated mice as compared to the garcinol group (Figure 6D). From protein expression analysis, as compared with the control group, the Lp(a) group showed an increased expression level of TNF-a, IL-6, CRP, NF-kB, and phosphorylation of CamKII/ERK/p38 MAPK medicated by α7-nAChR while the garcinol treatment group exhibited effects of attenuation (Figure 6E). The relative expression of miRs in blood isolated from the garcinol-treated group showed a higher level of miR-205 in comparison to that from the control and Lp(a) treatment group (Figure 6F). ELISA analysis results compared to the treatment group showed the significant reduction in the level of hemodynamic and cardiac function markers clusterin, endothelin-1 and troponin I (Figure 6G–I).

## 4. Discussion

We found that garcinol suppressed Lp(a)/α7-nAChR-mediated cardiomyocyte apoptosis and inflammation in the human cardiomyocyte AC16 cells and mouse models of isoproterenol-induced AMI through inhibition of IGF2R, p-GSK-3β/NF-κB, p-CamKII/p-ERK/p-p38 MAPK, RhoA-GTP signaling pathway, and reduced the production of IL-6, TNF-α, and CRP. The increased expression of miR-205 to inhibit CHRNA7 by garcinol played a key role in the mechanism by which garcinol prevents post-infarct cardiomyocyte apoptosis and inflammation.

Although oxidized, but not native, low-density lipoprotein causes apoptosis of different types of cell [31], the cytotoxic effect of Lp(a) remains undefined. We demonstrated that the exposure of human ventricular cardiomyocyte AC16 cells to Lp(a) increased oxidative stress, reduced the cell viability and caused the cell to undergo apoptosis. It has been recognized for decades that Lp(a) exerts its biological functions, at least in part, through the nicotinic receptors, among which activation of α7-nAChR enhances atherosclerosis in mice fed with a high fat-enriched diet [32]. While plasma concentrations of Lp(a) are reported to rise acutely under pathological challenge such as after myocardial infarction and percutaneous coronary intervention [33], it has been demonstrated that prolonged exposure to high circulating apolipoprotein(a) levels would render the vascular smooth muscle cells more contractile via the RhoA/ROCK-mediated mechanism [33]. Our finding that the downstream effector pathway by which Lp(a) activated cardiomyocytes relied on the α7-nAChR-dependent activation of p38 MAPK is consistent with the effect of α7-nAChR in dendritic cells [34]. The α7-nAChR agonists have been shown to trigger the activation of CamKII, leading to sequential activation of ERK and p38 MPAK in neuroblastic cells [35]. In our study, the increased expression of α7-nAChR by Lp(a) triggered a pro-inflammatory circuit, such as activation of RhoA-GTP, NF-kB, IL-6, TNF-α, and CRP. We further demonstrated that α7-nAChR-mediated phosphorylated activation of CamKII/ERK/p38 MAPK enhanced cardiomyocyte apoptosis, suggesting potential use of α7-nAChR as a cardiomyocyte apoptotic marker.

Cardiomyocytes express relatively high levels of IGF2R, which normally acts to suppress cell growth [36]. We found that Lp(a) increased the expression of IGF2R, suggesting Lp(a) as a negative regulator of post-infarct cardiomyocyte growth. In general, active GSK-3β is pro-inflammatory, and inhibited (phosphorylated) GSK3β predominantly ameliorates inflammation [25]. Furthermore, GSK-3β has been reported to both activate and inhibit NFκB depending on cell- and stimulus-selective interactions [37], as overexpression/activation is detrimental but sustained inhibition could be detrimental too. Notably, phosphorylation (inhibition) of GSK-3β exacerbates ischemic myocardial injury [38]. Our finding that garcinol decreased phosphorylation (inhibition) of GSK-3β in Lp(a)-related post-infarct cardiomyocyte apoptosis and inflammation is consistent with a previous study showing that the GSK-3β knockout mice after AMI display better-preserved heart function, reduced LV remodeling, and apoptosis [39]. Moreover, in line with a previous study [24], we demonstrated that TNF-α signaling simultaneously activated both proapoptotic p38 MAPK and caspase-3, along with canonical NF-κB/IκBα/p65 signaling. Further preclinical studies with conditional loss of function mouse models is warranted before TNF-α/p-GSK-3β/NF-κB inhibition could be considered as a therapy for post-infarct cardiomyocyte apoptosis.

Cardiomyocyte apoptosis is a necessary form of cell death in ischemia-reperfusion injury or re-oxygenation injury when the blood supply returns to tissue after a period of ischemia, with apoptotic rates of 2–12% in the border zone of human AMI [40,41]. While such a dramatic loss of viable tissue can have a disastrous effect on the geometry and function of the left ventricle, AMI-associated heart failure would be classified as one of the clinical disorders where apoptosis should be actively antagonized to limit cell loss [42]. In the development of anti-apoptosis strategies for heart failure, cell therapy to replace functional cardiomyocytes for AMI appears to have no apparent clinical benefit [43]. On the other hand, inhibiting cell death of pre-existing muscle cells may be promising. Anti-TNF-α therapy, however, provided no evidence of clinical benefit to heart failure patients [44]. In our study, when garcinol (2.5 μM) was introduced to Lp(a) treated AC16 ventricular cardiomyocytes, it decreased the effect of Lp(a) and protected cardiomyocytes from undergoing apoptosis. Furthermore, garcinol significantly and dose-dependently inhibited the expression of α7-nAChR, resulting in inhibition of α7-nAChR mediated phosphorylation activation of p-38/CamKII/ERK and ROCK activity, suggesting new mechanistic insights of this natural anti-apoptosis compound.

Consistent with previous studies [27,45], Lp(a) upregulates the expression of adhesion molecules, especially VCAM-1, ICAM-1, and E-selectin, in the inflammatory status, such as AMI. Furthermore, our cell-based ELISA demonstrated that garcinol reduced the Lp(a)/α7-nAChR-induced expression of VCAM-1, ICAM-1, and E-selectin and proinflammatory cytokines (IL-6, TNF-a, CRP, and NF-κB). An antagonist of p38 MAPK, SB203580, had a similar inhibitory effect of garcinol on Lp(a)-induced inflammation, suggesting that garcinol exerts the anti-inflammatory effects via α7-nAChR. In addition, the expression of F-actin, which is essential for cell stability [46], was increased upon garcinol treatment in comparison to Lp(a) and SB203580 in cardiomyocytes.

While miRNAs control cardiomyocyte cell death and proliferation in the infarcted heart, acute inhibition, or overexpression of miRNAs after AMI might be useful to limit tissue injury [47]. The mir-Target prediction in our study showed that the 3′UTR sites of the CHRNA7 gene, encoding α7-nAChR, were effectively targeted by miR-205. We also demonstrated for the first time that the suppression of α7-nAChR by garcinol both in vivo and in vitro, is associated with the significant up-regulation of miR-205 expression in cardiomyocytes. Moreover, exogenous injection of garcinol effectively alleviates AMI-related pathological changes in the myocardium, reduces myocardial apoptosis, decreases the expression of pro-inflammatory cytokines such as, TNF-α, IL-6, CRP, NF-kB, and reduces the α7-nAChR-mediated phosphorylation of CamKII-ERK-p38 MAPK.

Nevertheless, this study has several limitations. First, we did not precisely examine whether the multiple programmed cell death pathways under hypoxic conditions were necroptotic/apoptotic cell death, which represents a limitation of this study, particularly for elucidating the underlying mechanisms of miRNA 205-mediated cardiomyocyte cell survival effects. Further studies using necroptosis and apoptosis inhibitors specific for necroptosis and apoptosis factors will be helpful in identifying the exact cell death type of Lp(a)/α7-nAChR-mediated responses.

## 5. Conclusions

Garcinol significantly decreased Lp(a)/α7-nAChR-mediated cardiomyocyte apoptosis and inflammation in the cardiomyocyte AC16 cells through inhibition of IGF2R, p-GSK-3β/NF-κB, p-CamKII/p-ERK/p-p38 MAPK, RhoA-GTP signaling pathway, and reduced the production of IL-6, TNF-α, and CRP (Figure 7). The above phenomenon was observed in mouse models of isoproterenol-induced AMI. In addition, garcinol increased the expression of miR-205 to inhibit CHRNA7, suggesting that garcinol could be a potential dietary phytochemical candidate for AMI patients, and miR-205 may be a potential therapeutic target in post-infarct cardiomyocyte apoptosis and inflammation in the hopes of inhibiting cell death to avert, or at least prolong, the degeneration toward symptomatic heart failure.

## Figures and Tables

**Figure 1 antioxidants-10-00461-f001:**
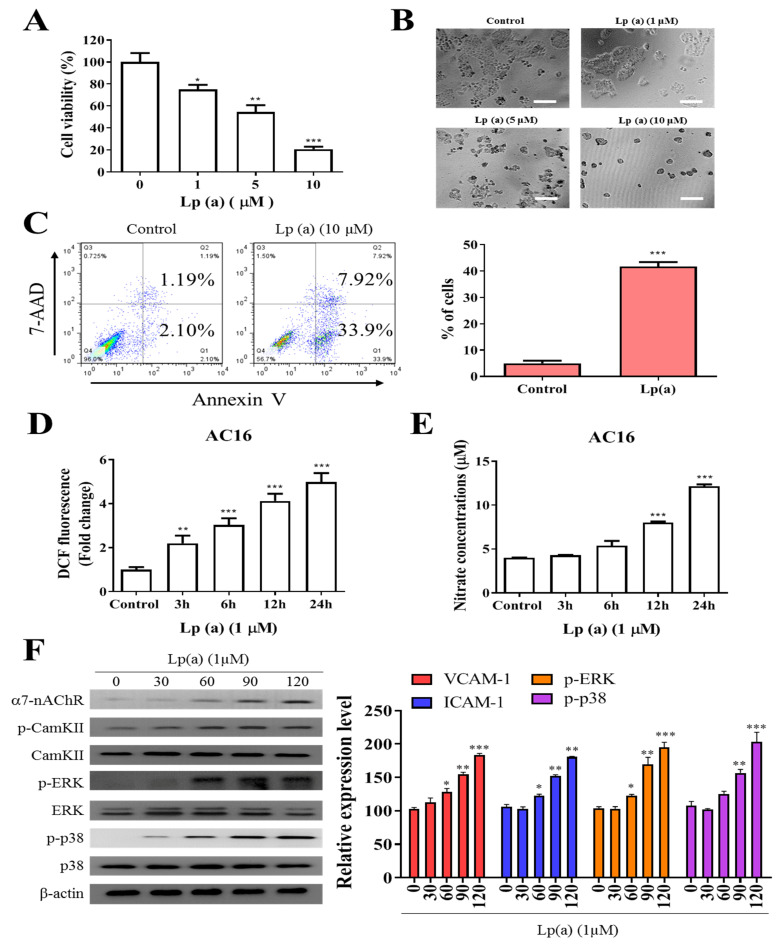
Effect of Lp(a) on reactive oxygen species (ROS) production and α7-nAChR (nicotinic acetylcholine receptor)-mediated phosphorylation in Ventricular cardiomyocyte AC16 cells. (**A**) Cell viability was significantly reduced in 5 and 10 μM Lp(a)-treated groups compared to control cells detected for 5 and 10 μM Lp(a)-treated groups compared to control cells. (**B**) Lp(a)-induced alterations in morphology after exposure to different concentration of Lp(a), (**C**) Lp(a)-induced apoptosis was detected using Annexin V-PE/7-AAD dual staining. (**D**) The cellular redox status was determined using 6-carboxy-2′,7′-dichlorodihydrofluorescein diacetate (carboxy-H2DCFDA). Serum-starved A16 cells was incubated with carboxy-H2DCFDA and treated with Lp(a) (1 μM) for the indicated time periods, and the fluorescence intensity was quantitated. (**E**) Serum-starved cells were treated with Lp(a) (1 μM) for the indicated time periods. The production of NO was determined by measuring nitrate concentrations. (**F**) Western blot analysis showed the concurrent changes in the expression of α7-nAChR and phosphorylation status of calmodulin-dependent kinase II (CamKII), p38 mitogen-activated protein kinase (MAPK) and extracellular signal-regulated kinase (ERK). * *p* < 0.05, ** *p* < 0.01, *** *p* < 0.001.

**Figure 2 antioxidants-10-00461-f002:**
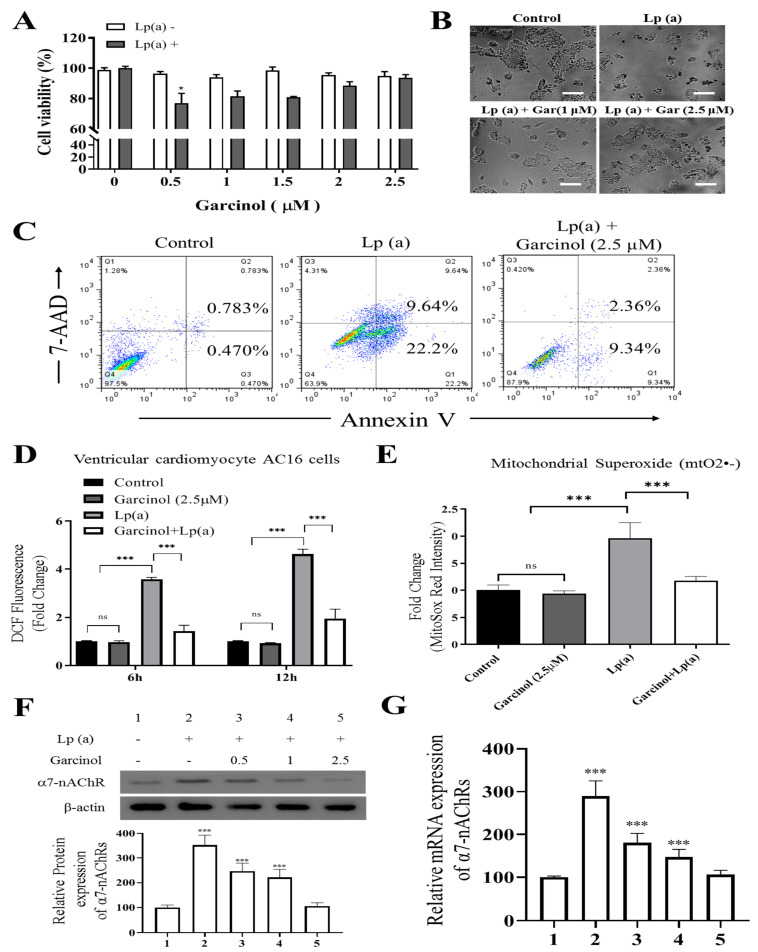
Garcinol reduced Lp(a)-induced reactive oxygen species (ROS) production and cytotoxicity in Ventricular cardiomyocyte AC16 cells. (**A**) Different concentrations of garcinol did not affect the viability of cardiomyocytes, but garcinol inhibited the cell cytotoxicity induced by Lp(a) in concentration-dependent manners. (**B**) Reduced alterations in morphology of Lp(a)-treated cardiomyocytes with simultaneous garcinol treatment. scale bars = 20 μm. (**C**) Garcinol inhibited Lp(a)-induced apoptosis in AC16 cell. Annexin V-PE/7-AAD dual staining was to detect apoptosis. (**D**) Garcinol suppressed Lp(a)-induced ROS production in human ventricular cardiomyocyte AC16 cells. AC16 were exposed to Lp(a) (1 μM), garcinol (2.5 μM) or Lp(a) (1 μM) with 6 h pretreatment with garcinol (2.5 μM). Six and 24 h after stimulation, AC16 cells were then incubated with DCFH-DA (20, 70-dichlorodihydrofluorescein diacetate), and the level of ROS production was detected by FACStar flow cytometer. (**E**) Ventricular cardiomyocyte AC16 cells were pretreated with Lp(a) (1 μM), garcinol (2.5 μM) or Lp(a) (1 μM) and mitochondrial superoxide mtO2^•−^ production-specific dye, MitoSOX^TM^ Red, before flow exposure. (**F**) The protein and (**G**) mRNA expression levels of α7-nAChR were significantly decreased by garcinol in a dose-dependent manner. * *p* < 0.05, ** *p* < 0.01, *** *p* < 0.001.

**Figure 3 antioxidants-10-00461-f003:**
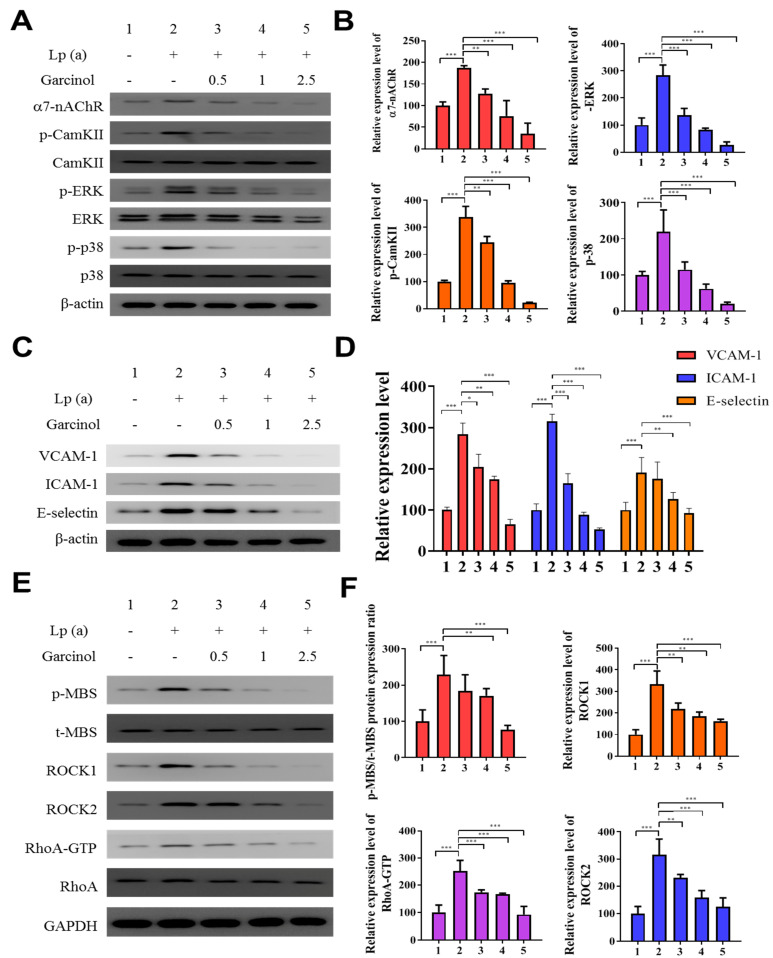
Garcinol inhibited the α7-nAChR-mediated phosphorylation and expression of adhesion molecules as well as RhoA-GTP activity in cardiomyocyte AC16 cells. (**A**) Western blot analysis showed that garcinol dose-dependently decreased the Lp(a)-induced activation of α7-nAChR and phosphorylation of CamKII, p38 MAPK and ERK. (**B**) Densitometry analysis of the immunoblots, shown as bar graphs normalized to β-actin. (**C**) Representative image of Western blot and (**D**) quantitative real-time polymerase chain reaction (qRT-PCR) mRNA analysis showing the dose-dependent reduction of Lp(a)-induced vascular cell adhesion molecule 1 (VCAM-1), intercellular adhesion molecule 1 (ICAM-1), and E-selectin expression by garcinol. (**E**) and (**F**) The Lp(a)-induced activation of the RhoA-GTP/Rho-kinases (ROCK1, ROCK2) were dose-dependently decreased by garcinol. * *p* < 0.05, ** *p* < 0.01, *** *p* < 0.001.

**Figure 4 antioxidants-10-00461-f004:**
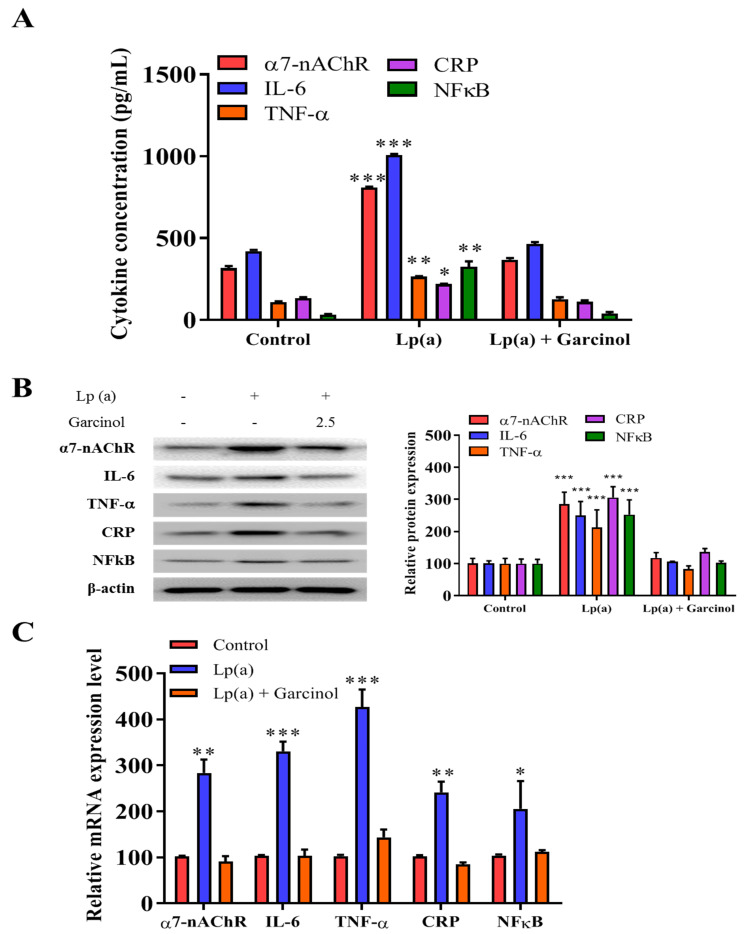
Garcinol inhibited Lp(a)-induced activation of α7-nAChR/p38 MAPK/NFkB in cardiomyocyte AC16 cells. (**A**) Enzyme-linked immunosorbent assay (ELISA) result showed that garcinol significantly decreased the Lp(a)-induced expression of α7-nAChR and proinflammatory cytokines (interleukin-6 (IL-6), tumor necrosis factor (TNF)-α, C-reactive protein (CRP), and NFkB). (**B**) Protein and (**C**) mRNA expression of α7-nAChR, IL-6, TNF-α, CRP, and NFkB were significantly reduced by garcinol. * *p* < 0.05, ** *p* < 0.01, *** *p* < 0.001.

**Figure 5 antioxidants-10-00461-f005:**
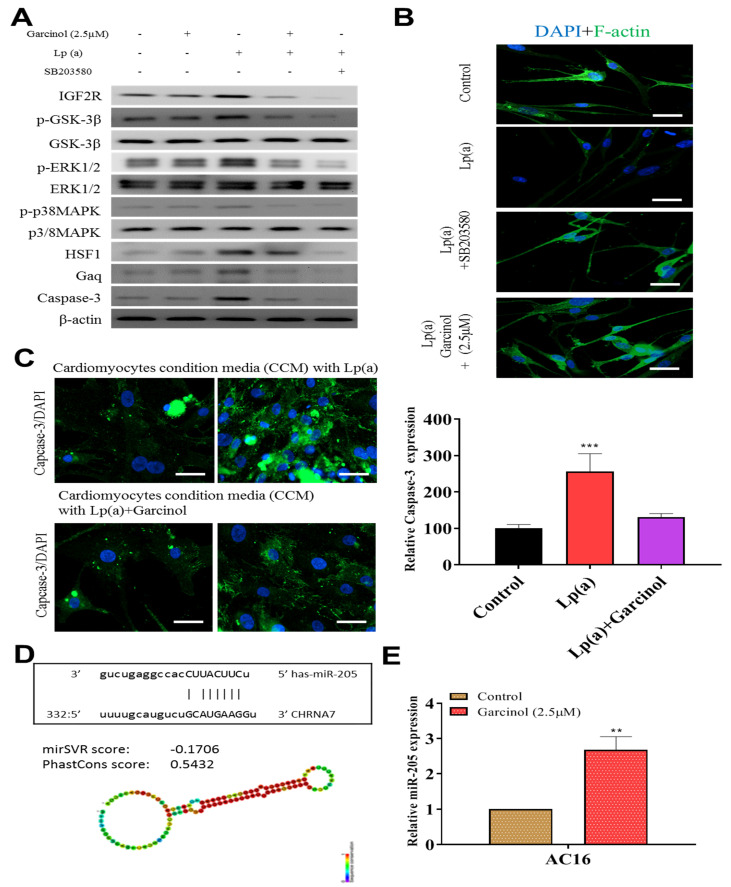
Garcinol prevented apoptosis and inhibited insulin-like growth factor-II receptor (IGF2R) through inactivation of phosphorylation of GSK-3β/ERK/p38 MAPK in ventricular cardiomyocyte AC16 cells. (**A**) Western blot analysis revealed that Lp(a) upregulated the expression of MAPK signaling-related proteins including p-GSK-3β, p-ERK1/2 and p-p38 MAPK. Garcinol significantly inhibited the Lp(a)-induced phosphorylation of GSK-3β, ERK1/2 and p38 MAPK, but had no obvious effect on total ERK1/2 and p38 MAPK expression. (**B**) In the immunofluorescence analysis of F-actin polymerization, nuclei and F-actin were stained, respectively, with 4′,6-diamidino-2-phenylindole (DAPI, blue) and rhodamine-phalloidin (red orange). An overlay of the two fluorescent signals was shown (scale bars = 20 μm). (**C**) Left panel: representative images of caspase-3 activation in AC16 cells treated with cardiomyocytes condition media in the presence of Lp(a) for 24 h with and without garcinol. Right panel: quantification of relative caspase-3 activation. A substantial reduction in caspase activation, suggesting less apoptosis, was observed in garcinol treatment in comparison to Lp(a) only treatment. (**D**) The mir-Target prediction showed the 3′UTR sites of CHRNA7 targeted by miR-205. (**E**) qPCR analysis showed miR-205 expression was significantly upregulated by garcinol in comparison to control. * *p* < 0.05, ** *p* < 0.01, *** *p* < 0.001.

**Figure 6 antioxidants-10-00461-f006:**
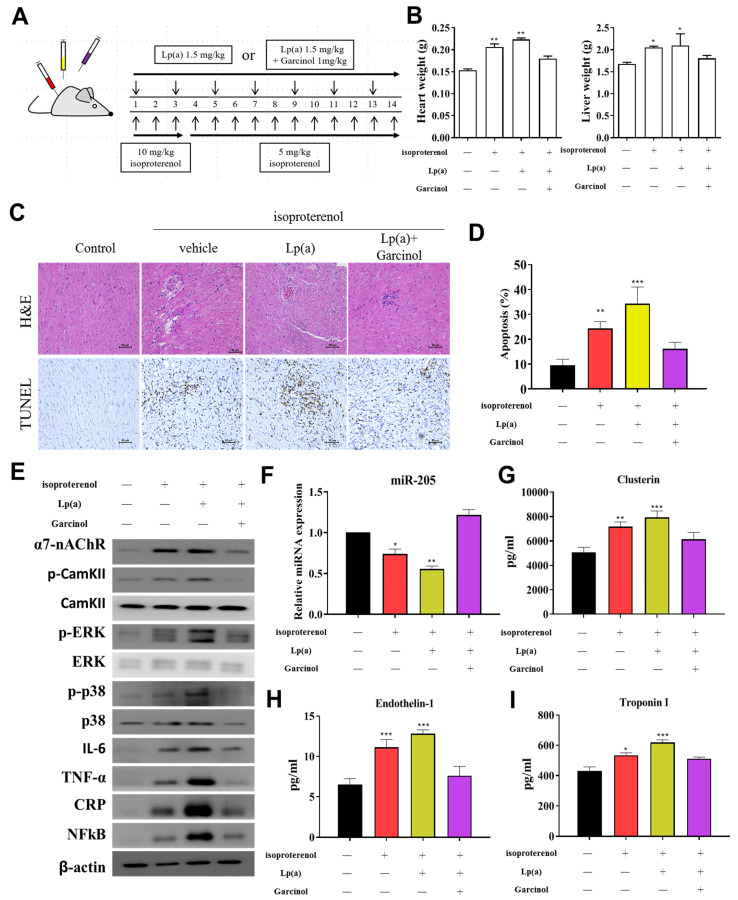
Garcinol suppressed Lp(a)-induced myocardial apoptosis and inflammation in a mouse model of myocardial infarction. (**A**) The protocol for the induction of myocardial infarction in C57/B6 mice. (**B**) The heart and liver weight of the control group, and Lp(a) treated mice as compared to the garcinol treatment in mice. (**C**) H&E and transferase dUTP nick end labeling (TUNEL) staining results of myocardium in the sham, Lp(a) and garcinol groups. (**D**) Quantitative analysis of percentage apoptosis. (**E**) Western blot showed a decrease in the expression of pro-inflammatory cytokines TNF-α, IL-6, CRP, NFκB, and phosphorylated CamKII/ERK/p38 MAPK by garcinol. β-actin was used as the loading control. (**F**) RT-qPCR analysis indicated that the relative miR-205 expression was significantly higher in mice treated with garcinol in comparison to Lp(a) group. (**G**–**I**) Enzyme-linked immunosorbent assay (ELISA) analysis results as compared to the treatment group, the significant reduction in the level of hemodynamic and cardiac function markers clusterin, endothelin-1 and troponin I. * *p* < 0.05, ** *p* < 0.01, *** *p* < 0.001.

**Figure 7 antioxidants-10-00461-f007:**
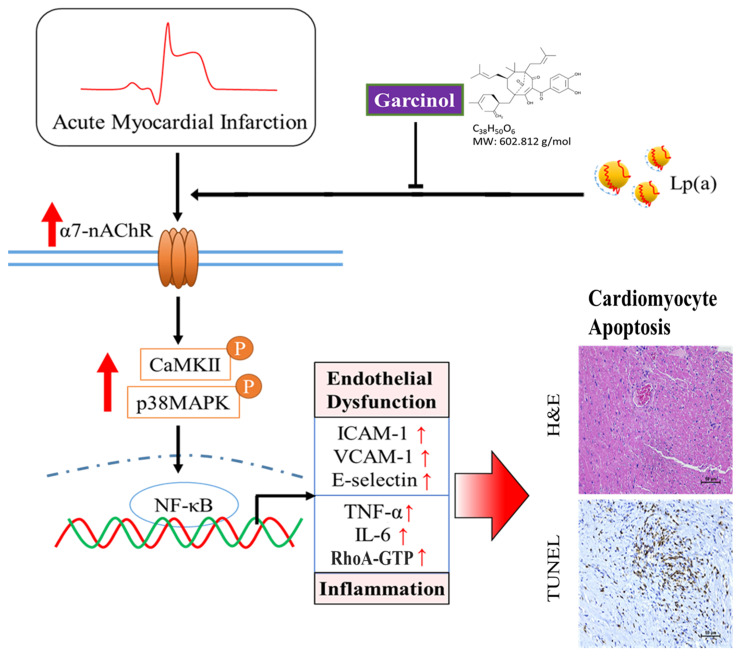
Garcinol significantly decreased the Lp(a)/α7-nAChR-mediated protein expression levels of inflammatory cytokines and phosphorylation activation of CamKII/p38 MAPK signaling, resulting in reduced post-infarct cardiomyocyte apoptosis.

## Data Availability

The datasets used and analyzed in the current study are publicly accessible as indicated in the manuscript.

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
