# Peer review of "Garcinol Attenuates Lipoprotein(a)-Induced Oxidative Stress and Inflammatory Cytokine Production in Ventricular Cardiomyocyte through α7-Nicotinic Acetylcholine Receptor-Mediated Inhibition of the p38 MAPK and NF-κB Signaling Pathways"

_antioxidants, 2021, doi:10.3390/antiox10030461_

Round 1

Reviewer 1 Report

The paper reported interesting findings about the cardioprotective effect of Garcinol. Garcinol, a polyisoprenylated benzophenone, has traditionally been extensively used for its antioxidant and anti-inflammatory properties. Several in vitro and in vivo studies have illustrated the potential therapeutic efficiency of garcinol in management of different malignancies. It mainly acts as an inhibitor of cellular processes via regulation of transcription factors NF-κB and JAK/STAT3 in tumor cells and have been demonstrated to effectively inhibit growth of malignant cell population. Less is known about is cardioprotective role following myocardial infarction.

The overall of this aim is to understand if Garcinol attenuates the toxic effect of Lp(a) in cardiomyocytes and in a mouse model of isoproterenol-induced AMI.

The paper is well written and data clearly presented and discussed, however some experiments and control point are missing.

1) Why the author didn’t measure mitochondrial ROS? It has been demonstrated that mitochondria play an important role in myocardial I/R injury as disturbance of mitochondrial dynamics, especially excessive mitochondrial fission, is a predominant cause of cardiac dysfunction.

2) Did the authors control the effect of Garcinol treatment alone (not in combination with Lp(a)?

3) What kind of diet did the mice follow?

4) In the animal experiments did the author control:

-Effects of garcinol on hemodynamic markers and cardiac function in isoproterenol-treated mice

-Effects of garcinol on heart and liver weight in isoproterenol-treated mice

Minor points

Lanes 265-269: what did the authors mean by Garcinol…making the cell healthier?

Figure 2B is not clear, please use a different magnification

Author Response

We thank the reviewer for carefully reading our manuscript and providing valuable comments. We accordingly response the questions raised by the Reviewer as follows:

Comments and Suggestions for Authors:

The paper reported interesting findings about the cardioprotective effect of Garcinol. Garcinol, a polyisoprenylated benzophenone, has traditionally been extensively used for its antioxidant and anti-inflammatory properties. Several in vitro and in vivo studies have illustrated the potential therapeutic efficiency of garcinol in management of different malignancies. It mainly acts as an inhibitor of cellular processes via regulation of transcription factors NF-κB and JAK/STAT3 in tumor cells and have been demonstrated to effectively inhibit growth of malignant cell population. Less is known about is cardioprotective role following myocardial infarction.

The overall of this aim is to understand if Garcinol attenuates the toxic effect of Lp(a) in cardiomyocytes and in a mouse model of isoproterenol-induced AMI. The paper is well written, and data clearly presented and discussed, however some experiments and control point are missing.

A1: We sincerely thank the reviewer for the time taken to review our work and the important critiques or suggestions made regarding same.

Response to Reviewers:

Reviewer #1 (Comments to the Author):

Q1: 1)      Why the author didn’t measure mitochondrial ROS? It has been demonstrated that mitochondria play an important role in myocardial I/R injury as disturbance of mitochondrial dynamics, especially excessive mitochondrial fission, is a predominant cause of cardiac dysfunction.

A1: We sincerely thank the reviewer for this insightful suggestion. We have now included mitochondrial ROS and determined the mitochondrial superoxide mtO2•-in updated Figure 2E.

Please kindly see our revised results section.

3.2   Garcinol prevents the apoptotic cell death induced by Lp(a)

To elucidate whether Garcinol treatment inhibited cell cytotoxicity caused by Lp(a), we performed cell viability assay by Cell Counting Kit-8 and annexin V/7-AAD apoptosis assay of AC16 cells, sensitized by 1 µM Lp(a), then treated with different concentration of garcinol (0.5-2.5 µM). Different concentrations of garcinol did not affect the viability of cardiomyocytes. Moreover, Garcinol inhibited the cell cytotoxicity induced by Lp(a) in concentration-dependent manners, with 0.5 µM being the lowest effective concentration (Figure 2A) and less vulnerable to Lp(a)-related cytotoxic effects, as shown in Figure 2B. Protective effects of garcinol against the cell apoptosis of AC16 cells were analyzed by flow cytometry analysis (Figure 2C), demonstrating that the 24 h incubation of AC16 cells with garcinol (1 μM) significantly reduced the apoptosis induced by Lp(a). Therefore, ROS level was quantified using flow cytometer 6 and 12 h after Lp(a) stimulation (10 μM) with or without garcinol pretreatment (2.5 μM) for 6 h. Similarly, Lp(a)-induced ROS production and mitochondrial superoxide mtO2•- production was markedly decreased by garcinol post-Lp(a) exposure (Figure 2D and 2E). To further confirm that garcinol was able to prevent α7-nAChR effects on AC16 cells, western blot and qRT-PCR analysis were used to evaluate the expression of α7-nAChR. Garcinol (0.5-2.5 μM) suppressed α7-nAChR expression both at protein and mRNA level dose-dependently (Figure 2F and 2G). Collectively, an exposure-response relationship exists between garcinol and the reduction in oxidative stress, in which both the amount and the duration of garcinol used modulate this association.

Please kindly see our revised Materials and methods section.

2.6   MitoSox staining assay

MitoSOX Red superoxide indicator (Invitrogen) is a fluorogenic dye that is selective for mtO2•- in live cells. It localizes into cellular mitochondria and is readily oxidized by superoxide, but not other sources of ROS or nitrogen species. The oxidation of the probe is prevented by superoxide dismutase and exhibits a bright red fluorescence upon binding to nucleic acids (excitation=emission maxima ¼ 510=580 nm). After Lp(a) exposure, ventricular cardiomyocyte AC16 cells were incubated with MitoSOX Red (3 μM) for 10 min at 37oC as indicated by the manufacturer’s instructions. The AC16 cells were collected by trypsinization and washed in PBS supplemented with 2% FBS. AC16 cells were fixed in 2% paraformaldehyde and suspended in PBS. Measurements were performed in duplicates using the BD flow cytometer (BD Biosciences). MitoSOX Red was excited at 488 nm, and the data were collected by a 575=26 nm (FL2) channel. The data were presented by histograms in terms of the mean intensity of MitoSOX fluorescence normalized to those of the Garcinol controls.

Please kindly see our revised Figure 2 Legends.

Figure 2: Garcinol reduced Lp(a)-induced reactive oxygen species (ROS) production and cytotoxicity in Ventricular cardiomyocyte AC16 cells.  (A) Different concentrations of garcinol did not affect the viability of cardiomyocytes, but Garcinol inhibited the cell cytotoxicity induced by Lp(a) in concentration-dependent manners. (B) Reduced alterations in morphology of Lp(a)-treated cardiomyocytes with simultaneous garcinol treatment. scale bars = 20 μm. (C) Garcinol inhibited Lp(a)-induced apoptosis in AC16 cell. Annexin V-PE/7-AAD dual staining was to detect apoptosis. (D) Garcinol suppressed Lp(a)-induced reactive oxygen species (ROS) production in human ventricular cardiomyocyte AC16 cells. AC16 were exposed to Lp(a) (1 μM), Garcinol (2.5 μM) or Lp(a) (1 μM) with 6 h pretreatment with Garcinol (2.5 μM). Six and 24 h after stimulation, AC16 cells were then incubated with DCFH-DA, and the level of ROS production was detected by FACStar flow cytometer. (E) Ventricular cardiomyocyte AC16 cells were pretreated with Lp(a) (1 μM), Garcinol (2.5 μM) or Lp(a) (1 μM) and mitochondrial superoxide mtO2•- production-specific dye, MitoSOX Red, before flow exposure. (F) The protein and (G) mRNA expression levels of α7-nAChR were significantly decreased by garcinol in a dose-dependent manner. *P <0.05, **P < 0.01, ***P <0.001.

Q2: Did the authors control the effect of Garcinol treatment alone (not in combination with Lp(a)?

A2: We sincerely thank the reviewer for this insightful suggestion. We have now included effect of Garcinol treatment alone in updated Figure 2 A.

Please kindly see our revised result section:

3.2   Garcinol prevents the apoptotic cell death induced by Lp(a)

To elucidate whether Garcinol treatment inhibited cell cytotoxicity caused by Lp(a), we performed cell viability assay by Cell Counting Kit-8 and annexin V/7-AAD apoptosis assay of AC16 cells, sensitized by 1 µM Lp(a), then treated with different concentration of garcinol (0.5-2.5 µM). Different concentrations of garcinol did not affect the viability of cardiomyocytes. Moreover, Garcinol inhibited the cell cytotoxicity induced by Lp(a) in concentration-dependent manners, with 0.5 µM being the lowest effective concentration (Figure 2A) and less vulnerable to Lp(a)-related cytotoxic effects, as shown in Figure 2B. Protective effects of garcinol against the cell apoptosis of AC16 cells were analyzed by flow cytometry analysis (Figure 2C), demonstrating that the 24 h incubation of AC16 cells with garcinol (1 μM) significantly reduced the apoptosis induced by Lp(a). Therefore, ROS level was quantified using flow cytometer 6 and 12 h after Lp(a) stimulation (10 μM) with or without garcinol pretreatment (2.5 μM) for 6 h. Similarly, Lp(a)-induced ROS production and mitochondrial superoxide mtO2•- production was markedly decreased by garcinol post-Lp(a) exposure (Figure 2D and 2E). To further confirm that garcinol was able to prevent α7-nAChR effects on AC16 cells, western blot and qRT-PCR analysis were used to evaluate the expression of α7-nAChR. Garcinol (0.5-2.5 μM) suppressed α7-nAChR expression both at protein and mRNA level dose-dependently (Figure 2F and 2G). Collectively, an exposure-response relationship exists between garcinol and the reduction in oxidative stress, in which both the amount and the duration of garcinol used modulate this association.

Please kindly see our revised Figure 2 Legends.

Figure 2: Garcinol reduced Lp(a)-induced reactive oxygen species (ROS) production and cytotoxicity in Ventricular cardiomyocyte AC16 cells.  (A) Different concentrations of garcinol did not affect the viability of cardiomyocytes, but Garcinol inhibited the cell cytotoxicity induced by Lp(a) in concentration-dependent manners. (B) Reduced alterations in morphology of Lp(a)-treated cardiomyocytes with simultaneous garcinol treatment. scale bars = 20 μm. (C) Garcinol inhibited Lp(a)-induced apoptosis in AC16 cell. Annexin V-PE/7-AAD dual staining was to detect apoptosis. (D) Garcinol suppressed Lp(a)-induced reactive oxygen species (ROS) production in human ventricular cardiomyocyte AC16 cells. AC16 were exposed to Lp(a) (1 μM), Garcinol (2.5 μM) or Lp(a) (1 μM) with 6 h pretreatment with Garcinol (2.5 μM). Six and 24 h after stimulation, AC16 cells were then incubated with DCFH-DA, and the level of ROS production was detected by FACStar flow cytometer. (E) Ventricular cardiomyocyte AC16 cells were pretreated with Lp(a) (1 μM), Garcinol (2.5 μM) or Lp(a) (1 μM) and mitochondrial superoxide mtO2•- production-specific dye, MitoSOX Red, before flow exposure. (F) The protein and (G) mRNA expression levels of α7-nAChR were significantly decreased by garcinol in a dose-dependent manner. *P <0.05, **P < 0.01, ***P <0.001.

Q3: What kind of diet did the mice follow?

A3: We sincerely thank the reviewer for this insightful suggestion. Please kindly see our revised Materials and methods section.

2.12 AMI mouse model studies

The animal study protocol was approved by the Animal Care and User Committee at Taipei Medical University (Affidavit of Approval of Animal Use Protocol # Taipei Medical University- LAC-2018-0573) consistent with the U.S. National Institutes of Health Guide for the Care and Use of Laboratory Animals. Male C57Bl/6 mice (8-week-old) purchased from BioLASCO (BioLASCO Taiwan Co., Ltd. Taipei, Taiwan). Standard pellet diet and sufficient water were provided to mice and maintained under standard laboratory conditions (21 ± 2oC; 60–65% humidity) at 12/12 h light and dark cycle in a polycarbonate cage. The induction of AMI was performed in the Infarcted group, through subcutaneous administration of isoproterenol at a dose of 10 mg/kg for 3 days and 5 mg/kg for 11 days. Following the isoproterenol was injected, randomly placed into Lp(a) control (Lp(a) 1.5 mg/kg) or garcinol-treated (Lp(a) 1.5 mg/kg, garcinol 1 mg/kg, intraperitoneal (i.p.) injection) group. Post experiment, the mice were humanely sacrificed, and heart samples were collected for further comparative immunohistochemistry and western analyses. [29, 30].

Q4: In the animal experiments did the author control. A. Effects of garcinol on hemodynamic markers and cardiac function in isoproterenol-treated mice. B Effects of garcinol on heart and liver weight in isoproterenol-treated mice

A4: We sincerely thank the reviewer for this insightful suggestion. We have now included effect of garcinol on cardiac function markers and heart/liver weight in updated Figure 6.

Please kindly see our revised Materials and methods section.

2.8   Cell-based ELISA analysis

The protocol used for cell ELISA of IL-6, TNF-α, CRP, NFĸB, α7-nAChR, Clusterin, Endothelin-1 and Troponin I have been modified from that of Rothlein et al. [26] and Takami et al. [27]. The optical density of each well was determined using a microplate reader at 450 nm within 30 min.

Please kindly see our revised results section.

3.6   Garcinol treatment suppressed Lp(a) induced inflammation and cell damage in isoproterenol-induced AMI mice

After the successful induction of AMI in mice (Figure 6A), we investigated the potential of Garcinol in reversing the effect of Lp(a) in-vivo. The heart and liver weight of the control group, and Lp(a) treated mice as compared to the garcinol treatment in mice. Pretreatment with Garcinol significantly decreased heart weight and liver weight. (Figure 6B). H&E staining results showed AMI was alleviated for each group (Figure 6C). These data indicate that Garcinol pretreatment inhibited cardiac hypertrophy. Furthermore, the result of TUNEL assay showed that the number of apoptotic cardiomyocytes was increased in the control group and Lp(a) treated mice as compared to the garcinol group (Figure 6D). From protein expression analysis, as compared with the control group, the Lp(a) group showed an increased expression level of TNF-a, IL-6, CRP, NF-kB, and phosphorylation of CamKII/ERK/p38 MAPK medicated by α7-nAChR while garcinol treatment group exhibited effects of attenuation (Figure 6E). The relative expression of miR's in blood isolated from the garcinol-treated group showed a higher level of miR-205 in comparison to that from the control and Lp(a) treatment group (Figure 6F). ELISA analysis results as compared to the treatment group, the significant reduction in the level of hemodynamic and cardiac function markers Clusterin, endothelin-1 and troponin I (Figure 6G-6I).

Please kindly see our revised Figure 6 Legends.

Figure 6. Garcinol suppressed Lp(a)-induced myocardial apoptosis and inflammation in a mouse model of myocardial infarction. (A) The protocol for the induction of myocardial infarction in C57/B6 mice. (B) The heart and liver weight of the control group, and Lp(a) treated mice as compared to the garcinol treatment in mice. (C) H&E and TUNEL staining results of myocardium in the sham, Lp(a) and garcinol groups. (D) Quantitative analysis of percentage apoptosis. (E) Western blot showed a decrease in the expression of pro-inflammatory cytokines TNF-α, IL-6, CRP, NFκB, and phosphorylated CamKII/ERK/p38 MAPK by garcinol. β-actin was used as the loading control. (F) RT-qPCR analysis indicated that the relative miR-205 expression was significantly higher in mice treated with garcinol in comparison to Lp(a) group. (G-I) ELISA analysis results as compared to the treatment group, the significant reduction in the level of hemodynamic and cardiac function markers Clusterin, endothelin-1 and troponin I. *P <0.05, **P < 0.01, ***P <0.001.

Minor points

Lanes 265-269: what did the authors mean by Garcinol…making the cell healthier?

A5: We sincerely thank the reviewer for this insightful suggestion. Please kindly see our revised results section.

3.2   Garcinol prevents the apoptotic cell death induced by Lp(a)

To elucidate whether Garcinol treatment inhibited cell cytotoxicity caused by Lp(a), we performed cell viability assay by Cell Counting Kit-8 and annexin V/7-AAD apoptosis assay of AC16 cells, sensitized by 1 µM Lp(a), then treated with different concentration of garcinol (0.5-2.5 µM). Different concentrations of garcinol did not affect the viability of cardiomyocytes. Moreover, Garcinol inhibited the cell cytotoxicity induced by Lp(a) in concentration-dependent manners, with 0.5 µM being the lowest effective concentration (Figure 2A) and less vulnerable to Lp(a)-related cytotoxic effects, as shown in Figure 2B. Protective effects of garcinol against the cell apoptosis of AC16 cells were analyzed by flow cytometry analysis (Figure 2C), demonstrating that the 24 h incubation of AC16 cells with garcinol (1 μM) significantly reduced the apoptosis induced by Lp(a). Therefore, ROS level was quantified using flow cytometer 6 and 12 h after Lp(a) stimulation (10 μM) with or without garcinol pretreatment (2.5 μM) for 6 h. Similarly, Lp(a)-induced ROS production and mitochondrial superoxide mtO2•- production was markedly decreased by garcinol post-Lp(a) exposure (Figure 2D and 2E). To further confirm that garcinol was able to prevent α7-nAChR effects on AC16 cells, western blot and qRT-PCR analysis were used to evaluate the expression of α7-nAChR. Garcinol (0.5-2.5 μM) suppressed α7-nAChR expression both at protein and mRNA level dose-dependently (Figure 2F and 2G). Collectively, an exposure-response relationship exists between garcinol and the reduction in oxidative stress, in which both the amount and the duration of garcinol used modulate this association.

Figure 2B is not clear, please use a different magnification

A6. We appreciate the editor’s insightful comment. We already changed different magnification of updated Figure 2B.

Reviewer 2 Report

The authors examine the role of the p38 MapK/NFkb pathway in mediating the anti-inflammatory effects of garcinol in a cardiac cell and cardiac infarct model.

The experimental approach is fairly standard but the experiments are well done and thorough.  The conclusions are appropriate based on the data presented.

There is some concern with the westerns throughout that exposure times vary greatly for some markers.

Some minor points:

In figure 1 and 2 the fonts are too small in some panels (most notably in figures 1 C and 2C. where the small font size makes it impossible to read where the gates are set for the flow cytometry studies).

Figures through out are pixelated and probably need to be at a higher resolution for publication.

In figure 5 immunocytochemistry studies need to presented at a higher resolution. And again, for sizes are too small (especially in 5D)

Author Response

Response to Reviewers:

Reviewer #2 (Comments to the Author):

Q1: The authors examine the role of the p38 MapK/NFkb pathway in mediating the anti-inflammatory effects of garcinol in a cardiac cell and cardiac infarct model. The experimental approach is fairly standard, but the experiments are well done and thorough.  The conclusions are appropriate based on the data presented. There is some concern with the westerns throughout that exposure times vary greatly for some markers.

A1: We sincerely thank the reviewer for the time taken to review our work and the important critiques or suggestions made regarding same.

Q2: In figure 1 and 2 the fonts are too small in some panels (most notably in figures 1 C and 2C. where the small font size makes it impossible to read where the gates are set for the flow cytometry studies).

A2: We sincerely thank the reviewer for this insightful suggestion. We already correct of the figure 1 and 2 fonts in revised panels. Please kindly see our updated Figure 1 and Figure 2.

Please kindly see our revised Figure 1 and Figure 2 Legends.

Figure 1: Effect of Lp(a) on ROS production and α7-nAChR-mediated phosphorylation in Ventricular cardiomyocyte AC16 cells. (A) Cell viability was significantly reduced in 5 and 10 μM Lp(a)-treated groups compared to control cells detected for 5 and 10 μM Lp(a)-treated groups compared to control cells. (B) Lp(a)-induced alterations in morphology after exposure to different concentration of Lp(a), (C) Lp(a)-induced apoptosis was detected using Annexin V-PE/7-AAD dual staining. (D) The cellular redox status was determined using carboxy-H2DCFDA. Serum-starved A16 cells was incubated with carboxy-H2DCFDA and treated with Lp(a) (1 μM) for the indicated time periods, and the fluorescence intensity was quantitated. (E) Serum-starved cells were treated with Lp(a) (1 μM) for the indicated time periods. The production of NO was determined by measuring nitrate concentrations. (F) Western blot analysis showed the concurrent changes in the expression of α7-nAChR and phosphorylation status of CamKII, p38 MAPK and ERK. *P <0.05, **P < 0.01, ***P <0.001.

Figure 2: Garcinol reduced Lp(a)-induced reactive oxygen species (ROS) production and cytotoxicity in Ventricular cardiomyocyte AC16 cells.  (A) Different concentrations of garcinol did not affect the viability of cardiomyocytes, but Garcinol inhibited the cell cytotoxicity induced by Lp(a) in concentration-dependent manners. (B) Reduced alterations in morphology of Lp(a)-treated cardiomyocytes with simultaneous garcinol treatment. scale bars = 20 μm. (C) Garcinol inhibited Lp(a)-induced apoptosis in AC16 cell. Annexin V-PE/7-AAD dual staining was to detect apoptosis. (D) Garcinol suppressed Lp(a)-induced reactive oxygen species (ROS) production in human ventricular cardiomyocyte AC16 cells. AC16 were exposed to Lp(a) (1 μM), Garcinol (2.5 μM) or Lp(a) (1 μM) with 6 h pretreatment with Garcinol (2.5 μM). Six and 24 h after stimulation, AC16 cells were then incubated with DCFH-DA, and the level of ROS production was detected by FACStar flow cytometer. (E) Ventricular cardiomyocyte AC16 cells were pretreated with Lp(a) (1 μM), Garcinol (2.5 μM) or Lp(a) (1 μM) and mitochondrial superoxide mtO2•- production-specific dye, MitoSOX Red, before flow exposure. (F) The protein and (G) mRNA expression levels of α7-nAChR were significantly decreased by garcinol in a dose-dependent manner. *P <0.05, **P < 0.01, ***P <0.001.

Q3: Figures throughout are pixelated and probably need to be at a higher resolution for publication.

A3: We thank the reviewers for the reference and suggestions. We have provided the higher resolution of all revised Figures for publication.

Q4: In figure 5 immunocytochemistry studies need to present at a higher resolution. And again, for sizes are too small (especially in 5D).

A4: We thank the reviewers for the reference and suggestions. We have provided the higher resolution of revised Figure 5 for publication.

Please kindly see our revised Figure 5 Legends.

Figure 5: Garcinol prevented apoptosis and inhibited insulin-like growth factor-II receptor (IGF2R) through inactivation of phosphorylation of GSK-3β/ERK/p38 MAPK in Ventricular cardiomyocyte AC16 cells. (A) Western blot analysis revealed that Lp(a) upregulated the expression of MAPK signaling-related proteins including p-GSK-3β, p-ERK1/2 and p-p38 MAPK. Garcinol significantly inhibited the Lp(a)-induced phosphorylation of GSK-3β, ERK1/2 and p38 MAPK, but had no obvious effect on total ERK1/2 and p38 MAPK expression. (B) In the immunofluorescence analysis of F-actin polymerization, nuclei and F-actin were stained, respectively, with DAPI (blue) and rhodamine-phalloidin (red orange). An overlay of the two fluorescent signals was shown (scale bars = 20 μm). (C) Left panel: representative images of caspase-3 activation in AC16 cells treated with cardiomyocytes condition media in the presence of Lp(a) for 24h with and without garcinol. Right panel: quantification of relative caspase-3 activation. A substantial reduction in caspase activation, suggesting less apoptosis, was observed in garcinol treatment in comparison to Lp(a) only treatment. (D) The mir-Target prediction showed the 3'UTR sites of CHRNA7 targeted by miR-205. (E) qPCR analysis showed miR-205 expression was significantly upregulated by garcinol in comparison to control. *P <0.05, **P < 0.01, ***P <0.001.

Round 2

Reviewer 1 Report

The authors have satisfactorily responded to all my questions and made the necessary changes to the manuscript

Reviewer 2 Report

The authors have addressed all of the concerns that I previously raised